# Nanomaterials Application in Orthodontics

**DOI:** 10.3390/nano11020337

**Published:** 2021-01-28

**Authors:** Wojciech Zakrzewski, Maciej Dobrzynski, Wojciech Dobrzynski, Anna Zawadzka-Knefel, Mateusz Janecki, Karolina Kurek, Adam Lubojanski, Maria Szymonowicz, Zbigniew Rybak, Rafal J. Wiglusz

**Affiliations:** 1Department of Experimental Surgery and Biomaterial Research, Wroclaw Medical University, Bujwida 44, 50-345 Wroclaw, Poland; wojciech.zakrzewski@student.umed.wroc.pl (W.Z.); adam.lubojanski@student.umed.wroc.pl (A.L.); maria.szymonowicz@umed.wroc.pl (M.S.); zbigniew.rybak@umed.wroc.pl (Z.R.); 2Department of Pediatric Dentistry and Preclinical Dentistry, Wroclaw Medical University, Krakowska 26, 50-425 Wroclaw, Poland; 3Student Scientific Circle at the Department of Dental Materials, School of Medicine with the Division of Dentistry in Zabrze, Medical University of Silesia in Katowice, Akademicki Sq. 17, 41-902 Bytom, Poland; wojt.dobrzynski@wp.pl; 4Department of Conservative Dentistry and Endodontics Wroclaw Medical University, Krakowska 26, 50-425 Wroclaw, Poland; anna.zawadzka-knefel@umed.wroc.pl; 5Department of Maxillofacial Surgery, Mikulicz Radecki’s University Hospital, Borowska 213, 50-556 Wroclaw, Poland; matjanecki@gmail.com; 6Rajdent, Kozielewskiego 9, 42-200 Czestochowa, Poland; karolinakurek93@gmail.com; 7International Institute of Translational Medicine, Jesionowa 11 St., 55–124 Malin, Poland; 8Institute of Low Temperature and Structure Research, Polish Academy of Sciences, Okolna 2, 50-422 Wroclaw, Poland

**Keywords:** nanomaterials, orthodontics, brackets, wires, antimicrobial effect

## Abstract

Nanotechnology has gained importance in recent years due to its ability to enhance material properties, including antimicrobial characteristics. Nanotechnology is applicable in various aspects of orthodontics. This scientific work focuses on the concept of nanotechnology and its applications in the field of orthodontics, including, among others, enhancement of antimicrobial characteristics of orthodontic resins, leading to reduction of enamel demineralization or control of friction force during orthodontic movement. The latter one enables effective orthodontic treatment while using less force. Emphasis is put on antimicrobial and mechanical characteristics of nanomaterials during orthodontic treatment. The manuscript sums up the current knowledge about nanomaterials’ influence on orthodontic appliances.

## 1. Introduction

Nanomaterials are widely used in modern clinical dentistry. They improve various properties, such as antimicrobial properties, durability of materials. These particles do not exceed 100 nm, due to they obtain a better ratio between the surface and mass. The larger the surface area of the material, the greater its reactivity. It is also easier to absorb them in the body, which can also result in high cytotoxicity [1]. Nanomaterials are used in many areas of dentistry, such as conservative dentistry, endodontics, oral, and maxillofacial surgery, periodontics, orthodontics, and prosthetics [2]. Orthodontics is a branch of dentistry dealing with the improvement of occlusal conditions and facial aesthetics in both children and adults. In cooperation with other specialists (such as dental surgeons, maxillofacial surgeons, periodontists), the orthodontist is able to significantly improve the patient’s quality of life [3]. Nanotechnology is used, among others, in brackets, archives, elastomeric ligatures, orthodontic adhesives. Improving the microbicidal properties, reducing friction and increasing the strength of the material are some of the advantages. However, a significant problem is the potential cytotoxicity of nanomaterials, therefore further research is needed [2].

The prolonged process of wearing orthodontic braces results in increased accumulation of dental plaque and eventually results in a greater risk of caries. Its development is generally associated with the activity of cariogenous bacteria due to prolonged dental plaque accumulation on teeth surfaces, deficiencies, avitaminosis, and diet. The demineralization process that starts the caries is called a white spot lesion (WSL), meaning, that decalcification of enamel surfaces adjacent to the orthodontic appliances is directly associated with orthodontic treatment [4]. Several studies confirm the accelerated accumulation of WLS in orthodontic treatments. Such tendency creates clinical problems leading to unacceptable esthetic alterations that, in some cases, might lead to conservative, restorative treatment. Research shows that more plaque can accumulate around composites compared to other restorative materials, which results in an increased percentage of secondary caries [5]. Moreover, resin composites do not have bacteriostatic properties.

Promising results in the prevention of pathological changes associated with orthodontic treatment are obtained through the use of nanotechnology. According to the European Commission states that: “Nanomaterial is defined as a natural, incidental, or manufactured material containing particles, in an unbound state or as an aggregate or as an agglomerate and where, for 50% or more of the particles in the number size distribution, one or more external dimensions is in the size range 1–100 nm. In specific cases and where warranted by concerns for the environment, health, safety, or competitiveness the number size distribution threshold of 50% may be replaced by a threshold between 1% and 50%” [6]. Implication of nanotechnology is beneficial to humans, it has been broadly used in the modern dentistry in in restorative dentistry as an additive nanoparticle with remineralizing properties in composite resins, dental adhesives, oral care products, in the control of bacterial biofilm as an antibacterial and antimineralizing additive in dental hygiene products such as toothpaste, mouth rinses, and composite resins. Nanotechnology is useful in the diagnosis of malignant and precancerous cavity diseases, periodontal diseases, and is also used in implantology—as a modification of the implant surface [7] and in the use of impression materials [8]. The development of technology gives better opportunities to both patient and orthodontist due to new physicochemical, mechanical and antibacterial properties of nanosized materials and can be used in coating orthodontic wires, elastomeric ligatures, and brackets, producing shape memory polymers and orthodontic bonding materials. Not only can we control biofilm formation, reduce bacterial activity and act anticariogenic, but also, through the desired tooth movement, shorten the treatment time.

There are many advantages in medicine of using nanotechnology; however, it creates many doubts regarding the safety for humans and the environment. Nanoparticles can easily penetrate tissues and can affect biological behaviors at different levels. It is necessary to conduct detailed research on the environmental and toxicological properties in order to assess the risk and lead a sustainable application of nanomaterials. The aim of this work was to describe and summarize the current use of nanoparticles and their antibacterial activity in orthodontics, including resin, brackets, and archwires.

## 2. Nano-Coatings in Orthodontic Archwires

Minimizing the frictional forces between the orthodontic wire and brackets has the potential to increase the desired tooth movement and thus shorten treatment time. In recent years, nanoparticles have been used as a component of dry lubricants. These solid-phase materials are capable of reducing the friction between two sliding surfaces without the need for a liquid medium. One of the many examples are Inorganic fullerene-like tungsten sulfide nanoparticles (IF-WS2) that are used as self-lubricating coatings for orthodontic stainless steel wires [9]. Friction tests simulating the performance of coated and uncoated wires were carried out on an Instron machine, scanning electron microscopy (SEM) and energy dispersive X-ray spectroscopy (EDS) analysis of the coated wires showed a clear impregnation of IF-WS2 nanoparticles in the Ni-P matrix.

Atomic force microscopy (AFM) was used as a tool to assess the surface roughness of stainless steel (SS), beta-titanium (β-Ti), and nickel-titanium(NiTi) wires [10]. The surface roughness measurement of the AFM method confirmed the fact that the roughness of the measures on the effectiveness of sliding mechanics, the corrosion behavior, and aesthetics of orthodontic arches. The influence of decontamination and clinical exposure on the modulus of elasticity, hardness and surface roughness of SS and NiTi arches, and AFM paper coupled with a nanoindenter were assessed [11]. The results of the AFM popularity assessment that the decontamination regimen and clinical exposure had no statistically significant effect on NiTi wires, but had a statistically significant effect on SS wires. In a diagnostic study, the clinical significance of statistical studies, analysis, and testing of the arch equipment on orthodontic movement is not predicted.

### 2.1. Nano Coatings Reducing Friction on Orthodontic Archwires

Orthodontic arches are used to generate biomechanical forces that are transmitted through the brackets to move the teeth and correct malocclusion, spacing, or crowding. They are also used for retention purposes, i.e., to keep the teeth in their current position. Currently, orthodontic arches are made of non-precious metal alloys. The most common types of wire are SS, NiTi, and β-Ti alloy wires. In the case of sliding mechanics, friction between the wire and the lock is one of the major factors influencing tooth movement. When one moving object makes contact with another, friction occurs on the contact surface, which causes resistance to the movement of the teeth. This frictional force is proportional to the force with which the contacting surfaces are pressed against each other and is governed by the interface surface characteristics (smooth/rough, chemically reactive/passive, or lubricant modified). Minimizing the frictional forces between the orthodontic wire and brackets will accelerate the desired tooth movement and thus shorten the treatment time.

NiTi substrates can be coated with cobalt and a layer of IF-WS2 nanoparticles using the electrodeposition method. The coated substrates showed friction reduction of up to 66% when compared to the uncoated ones. The results of such studies may have potential applications in reducing friction when using NiTi orthodontic wires. On the other hand, allergic reactions in patients with nickel sensitivity may be the disadvantage of introducing nickel into this type of coating. Therefore, the effect of such NiP coatings on stainless steel and NiTi wires should be assessed for biocompatibility in animal models and further human trials.

### 2.2. Delivering Nanoparticles from an Elastomeric Ligature

Elastomeric ligatures can serve as a support scaffold to deliver nanoparticles that can be anti-cariogenic or anti-inflammatory. They may also carry embedded antibiotic drug molecules. The release of anti-cariogenic fluoride from elastomeric ligatures has already been described in the literature [12,13]. Research has shown that fluoride release is characterized by an initial burst of fluoride in the first few days followed by a logarithmic fall. The whole process is effective against common enamel demineralization around the orthodontic bracket during treatment [14]. 

### 2.3. Shape Memory Polymers (SMP) in Orthodontics

In the last decade, there has been a growing interest in the production of aesthetic orthodontic wires to complement brackets in the color of the teeth. Shape memory polymers (SMPs) are materials that can remember equilibrium shapes and then manipulate and fix them into a temporary or dormant shape under certain temperature and stress conditions. They can later relax to their original, stress-free state under thermal, electrical, or environmental conditions. This relaxation is related to the elastic deformation stored in the previous manipulation. Recovery of SMP into equilibrium shape can be accompanied by an appropriate and prescribed force, useful for orthodontic tooth movement, or a macroscopic change in shape that is useful in ligation mechanisms. Due to the ability of SMP to have two shapes, these devices meet requirements unattainable by modern orthodontic materials, allowing the orthodontist to insert them into the patient’s mouth more easily and comfortably [15].

When placed in the oral cavity, these polymers can be activated by body temperature or light-activated photoactive nanoparticles thereby causing tooth movement. SMP orthodontic wires can provide an improvement over traditional orthodontic materials as they provide lighter, more consistent forces which, in turn, can cause less pain to patients. Also, SMP materials are transparent, stainable, and stain-resistant, providing the patient with a more aesthetic apparatus during treatment. High percent elongation of the SMP apparatus (up to about 300%) allows for the application of continuous forces over a large range of tooth movement, and thus, fewer patient visits [16,17]. Future directions of research on shape—nanocomposite polymers with memory for the production of aesthetic orthodontic wires may have interesting potential in the research of orthodontic biomaterials.

### 2.4. Control of Oral Biofilms during Orthodontic Treatment

Nanoparticles have a larger surface area to volume ratio (per unit mass) compared to non-nano scale particles, interacting more closely with microbial membranes and providing a much larger surface area for antimicrobial activity. In particular, metal nanoparticles with a size of 1–10 nm showed the highest biocidal effect on bacteria [18]. Silver has a long history of use in medicine as an antibacterial agent [19]. The antimicrobial properties of nanoparticles have been exploited through the mechanism of joining dental materials with nanoparticles or coating the surface with nanoparticles to prevent adhesion of microbes to reduce biofilm formation [20,21]. It was found that resin composites containing fillers implanted with silver ions that release silver ions have an antibacterial effect on oral *streptococci* [22].

Ahn et al. [16] compared an experimental composite adhesive (ECA) containing silica nanofillers and silver nanoparticles with two conventional composite adhesives and a resin-modified glass ionomer (RMGI) to investigate the surface characteristics, physical properties, and antimicrobial activity against cariogenic *streptococci*. The results suggest that the ECAs had rougher surfaces than conventional adhesives due to the addition of silver nanoparticles. Bacterial adhesion to ECA was lower than to traditional adhesives, which was not affected by saliva. Bacterial suspensions containing ECA show slower growth of bacteria than those containing conventional adhesives. There is no significant difference in the shear bond strength and fracture strength of the bond between ECA and conventional adhesives. 

## 3. Bracket Materials

The development of technology for the production of orthodontic materials and products provides better opportunities for patients with functional, health, and aesthetic results. It also improves the daily technical performance of the orthodontist. To perform their function properly, the brackets should have good biocompatibility, correct hardness, and strength, smooth archwire slot to reduce frictional resistance, smooth surface to reduce plaque deposition, should be precisely manufactured for each tooth, have high corrosion resistance, and ionic release [23]. 

Orthodontic braces are manufactured by three main methods which may be used in combination: Casting, injection molding, and milling from different types of material including metal, plastics, ceramics, and combinations. 

Among the compositions of metallic, we can distinguish stainless steel, non-nickel steel, low-nickel stainless, cobalt–chromium alloys, titanium, and its alloys, gold alloys, and platinum alloys [24].

Metallic materials and their alloys are characterized by high mechanical parameters, usually better than ceramics or polymers. The surface in contact with the wire should have a relatively high modulus of elasticity to minimize the disbursement of energy transmitted by the wire from inexpedient plastic deformation and difficult enough to minimize expenditure wear caused by the movement of the activated wire. On the other hand, the base of the bracket must be sufficiently deformable to facilitate removal during treatment completion [24].

Stainless steel is a metallic alloy commonly used in the production of orthodontics brackets, due to its low cost, higher modulus of elasticity, and good biomechanical properties [25]. It can be classified as austenitic, martensitic, ferritic, duplex (austenitic-ferritic), and precipitation-hardening. The most commonly used alloys in orthodontic brackets are 303, 304, 316, 317, 17-4 PH [26]. Conventional type 316 L austenitic stainless steel is composed of %wt: Iron: Balance, manganese: 2.0, chromium: 16–18, nickel: 10–14, molybdenum: 2–3, and traces of phosphorus, sulfur, and carbon [27]. Although this alloy works well in clinical use, signs of corrosion have been observed. 

17-4 PH alloy demonstrates improvement in corrosion resistance, frictional behavior, and cytotoxicity, more than austenitic stainless steels 303, 204, and 316/316 L [26]. Nickel stabilizes the austenitic phase; the anti-corrosive properties and the ductility are improved while the addition of chromium facilitates the formation of a passive anti-corrosion coating [28].

Although allergenic, cytotoxic, and mutagenic their content in the brackets is so small that their use is safe. Since the occurrence of adverse reactions, it was considered that exposure to these elements should be kept to a minimum. This resulted in the introduction of various non-nickel or very low nickel content stainless steel which is more resistant to corrosion and does not release nickel into the oral cavity. Compared to 316 L, Alloy 2205 is harder and less corrosive [29].

Titanium was used in the construction of brackets as a material with a proven lack of allergenicity and increased corrosion resistance. The many current dental and medical applications have made titanium the obvious choice of all the available components. 

Commercially pure titanium grade 4 and Ti-6Al-4V alloy are the most widely used types for manufacturing orthodontic brackets, The different methods of obtaining the brackets result in significant differences in physical, mechanical, and bulk material properties. Corrosion resistance is achieved due to the presence of a thin passive protective layer made of titanium oxide. This layer is more stable than its counterpart chrome oxide on stainless steel [30]. Gold-coated brackets were introduced as an alternative to steel and titanium brackets. They are plated with 300 micro inches of 24 karat gold, therefore, have significantly brighter appearance. Moreover, they have better mechanical properties compared with conventional brackets made of stainless steel alloys. Gold alloy brackets are introduced as highly anticorrosive and the first choice for patients allergic to nickel (Ni) [30]. Significant side effects have not been observed clinically. 

Additionally, nano-sized gold particles can be used on orthodontic appliances e.g., aligners, to increase its antibacterial activity, by preventing biofilm formation as can be seen in Figure 1. Both the gingiva and teeth are covered by aligners for almost the entire day, which is a risk factor for plaque accumulation. Gold particles also show positive biocompatibility both in vitro and in vivo. 

A suitable substitute for stainless steel brackets are those coated with a platinum layer. Platinum has been found as a material totally compatible in the oral environment. Its alloys are five times more resistant to abrasion than gold and compared to stainless steel, they have excellent corrosion resistance, a harder surface which reduces friction and improves the mechanics of sliding. As a combination of the platinum layer and the unique implantation process, a barrier has been created that protects against the diffusion of nickel, cobalt, and chromium. 

Similar electrochemical properties, including excellent corrosion resistance, to that of platinum brackets, are demonstrated by those made of cobalt chrome steel [31]. Regarding friction resistance, cobalt–chromium brackets are comparable, but have slightly less friction than stainless steel brackets when used with stainless steel wires; however, cobalt–chromium brackets offer more friction than titanium brackets with both stainless steel and beta-titanium wires [32].

Although metal brackets exhibit excellent mechanical properties and provide many clinical advantages the issue of aesthetics remains a challenge. Elements made from ceramics and plastics have been widely used in clinical orthodontics.

The first plastic brackets appeared in the early 1970s and were made of acrylic, then polycarbonate, but unfortunately problems related to them were quickly noticed. They had a tendency to water sorption, change color upon contact with the ultraviolet light and some food or drinks [33].

There has been observed an increased adhesion of pathogens like *Streptococcus mutans* and *Candida albicans*. In order to eliminate problems and improve their properties the following solutions are possible: Reinforcement with other materials such as ceramic or fiberglass fillers and/or metal slots, chemical modification of the polymer and alternative polymers for instance urethane dimethacrylate, high-density polyethylene, and EBP [34]. Research shows that compared to stainless steel brackets, plastic brackets are only suitable for clinical use if they have a metal slot [35].

An important issue is the biocompatibility of plastic materials, especially in terms of cytotoxic effects of particle- and fiber-reinforced polycarbonate orthodontic brackets in fibroblast and breast cancer cells through the activation of mitochondrial cell death mechanisms [36].

Although polycarbonate brackets with metal reinforced slots demonstrate a significantly lesser degree of deformation, followed by pure polyurethane, pure polycarbonate, and fiberglass reinforced polycarbonate brackets torque problems still exist. Ceramic reinforced polycarbonate brackets showed the highest deformation under torque stresses [37].

Polyoxymethylene brackets were found to be harder and less rough. Unfortunately, this material is also unattractive due to the opacity and milky color. Moreover, it appears to release some formaldehyde over time.

There is still a search for an ideal polymer that would combine the optical properties of translucency and the mechanical properties of stiffness, resistance to water absorption, and degradation. The introduction of new materials should ensure this does not release toxic compounds, in particular leaching of monomer Bis-GMA (bisphenol A-glycidyl methacrylate), TEGDMA (triethylene glycol dimethacrylate) [38]. The advantage of polymer brackets, as in the case of those from stainless steel, is the ease and safety of removing them from the tooth. 

Among the brackets ensuring excellent aesthetic and optimum stable properties, we also include those made of ceramics. Their advantages are high rigidity and abrasion resistance as well as biocompatibility, and they are free from discoloration. Ceramic brackets are usually composed of aluminum oxides. There are two varieties currently available polycrystalline and monocrystalline (Saphire) forms, depending on their method of production. Another category is the polycrystalline Zirconia which has been offered as an alternative to alumina ceramic [39]. Polycrystalline zirconia brackets have the greatest toughness amongst all ceramics however are very opaque and can exhibit intrinsic colors. The monocrystal alumina brackets, which are noticeably clearer and consequently more aesthetic, along with having higher strength, than the polycrystalline alumina brackets, show low fracture toughness, due to the lack of internal grain boundaries, the presence of pores, and machining damage from milling [26]. Ceramic materials have some disadvantages associated with iatrogenic enamel damage due to their hardness, bonding and debonding, Frictional resistance. Orthodontists may experience problems with bracket breakage and fracture resistance, particularly when trying the ligature or fracture from archwires forces.

## 4. Nanomaterials in Orthodontics

Nanomaterials versatility allows them to be used in many situations during orthodontic clinical treatment, as can be seen in Table 1. 

Friction is one of the major factors present during retraction or alignment of teeth during orthodontic treatment. One of the methods to overcome high friction is the application of higher forces during treatment. Such action can have one significant disadvantage- undesirable anchorage loss [54]. On the other hand, there are other methods of overcoming unwanted friction, including alteration of the bracket design or wire shape and size. At last, there is a possibility of nanoparticle coating addition. To benefit from the antibacterial properties of nanoparticles, there are two main strategies in orthodontics to reduce biofilm formation. One strategy focuses on coating the surface of orthodontic brackets or wires with nanoparticles [55]. The other is about combining nanoparticles with orthodontic adhesives or acrylic materials. The advantages of nanocomposite materials include excellent optical properties, easy handling, and excellent polishability [24]. Moreover, nanofillers can reduce the surface roughness of orthodontic adhesives, which is one of the most important factors in bacterial adhesion [25], as can be seen in Figure 2.

### 4.1. Silver Nanoparticles (AgNPs) Coating

Some studies have proposed silver nanoparticles as the most effective type of metal nanoparticles for preventing the growth of *Streptococcus mutans* [56]. Recently, silver nanoparticles (AgNPs) have been shown to be materials with excellent anti-microbial properties in a wide variety of microorganisms. In the orthodontic field, studies have incorporated AgNPs (17 nm) into orthodontic elastomeric modules, orthodontic brackets, and wires, and others, against a wide variety of bacterial species concluding that these orthodontic appliances with AgNPs could potentially combat the dental biofilm decreasing the incidence of dental enamel demineralization during and after the orthodontic treatments [57,58]. AgNPs can significantly inhibit the bacterial adherence of the *S. mutans* strain on the surfaces of the orthodontic bracket and wire appliances finding that the smaller AgNP samples demonstrated statistically to have the most important *S. mutans* antiadherence activities for orthodontic brackets and wires when compared to NiTi (nickel–titanium) and SS (stainless steel wires) [59]. It is also confirmed by several studies, that coverage of AgNPs in human dentin prevents biofilm formation on the surface of the dentin, together with bacterial growth inhibition [58,60,61]. In order for AGNPs to be a stable suspension able to limit the agglomeration, they should have zeta potential values ranging between +30 and −30 mV [62,63]. Bürgers et al. [64] confirms, that smaller AgNPs have the ability to release more silver ions, which promotes their antimicrobial effect, while the histological effect of AgNPs generally focuses on inhibition of microbial metabolism, leading to impaired production of extracellular polysaccharides and specific bacterial processes leading to its general dysfunction [65]. These studies confirm, that AgNP-coated brackets can help to decrease the spot lesions appearance during orthodontic treatment, and may be even useful in compromised patients with immune deficiency, diabetes, or elevated risk of endocarditis [66]. In addition to silver, many other nanoparticles like chitosan, copper, zinc, hydroxyapatite, and silicon dioxide can be added to composites in order to reduce bacterial activity and growth.

### 4.2. Chitosan

Chitosan is a naturally acquired polysaccharide that is formed by the deacetylation of chitin. It is a non-toxic, biodegradable, biocompatible, and has antibacterial properties [67], on *Agregatibacter actinomycetemcomitans*, *Porphyromonas Gingivalis,* and *Streptococcus mutans* [68,69]. Chitosan additionally has inhibiting action against fungi. This material’s application as an antibacterial chemical agent in mouthwashes is limited due to its reduced solubility in water. Nonetheless, its characteristics are highly desirable in dental materials. Chitosan could be maintained inside the materials in the oral cavity due to its insolubility in water. Histologically, inhibition is caused by inactivation of the enzyme, the substitution of lipopolysaccharides, metal ions, and formation of acidic polymer like teichoic acid. Chitosan, due to its low solubility and melting temperature, can be maintained in the oral cavity for a long period of time, unlike CHX which is released and disappears in the early phase.

### 4.3. Copper Oxide

It was proved by Yassaei et al. [70], that no significant difference was found between silver and copper oxide (CuO) nanoparticles, but it was noted that a curing time increased with the use of copper material when compared to the silver one. The former is cheaper and additionally both physically and chemically more stable than the latter. CuO nanoparticles affect *Streptococcus mutans* bacteria in a similar way as silver particles do [56]. It was confirmed in other studies [4], that copper and copper-zinc nanoparticles had a significant inhibitory effect on the studied microbes. According to other studies, CuO is able to decrease biofilm formation from 70 up to 80% [71]. Moreover, the similar results were achieved when CuO particles were incorporated into adhesive materials [72]. Additionally, nanoparticles like CuO can act as nano-fillers and enhance the shear bond strength of adhesive.

### 4.4. Nitrogen-Doped Titanium Dioxide (N-Doped TiO_2_) Brackets

The activation of N-doped TiO_2_ leads to the formation of OH. Free radicals, superoxide ions (O_2_), hydrogen peroxide (H_2_O_2_), and peroxyl radicals (HO_2_). These chemicals exert antimicrobial activity, also reacting with lipids, enzymes, and proteins. According to Poosti et al. [73], TiO_2_ nanoparticles of size 21 ± 5 nm can be blended to light cure orthodontic composite paste in 1, 2, and 3% and all these concentrations have similar antibacterial effects. Salehi et al. [74] proved, that nitrogen-doped TiO_2_ brackets have shown better antimicrobial activity when compared to the uncoated stainless steel brackets. Adding TiO_2_ to adhesives enhances its antibacterial activity without compromising its mechanical properties [75]. Nitrogen-doped TiO_2_ brackets were also reported to present antibacterial activity against normal oral pathogenic bacteria [76].

### 4.5. Zinc Oxide (ZnO)

It has been observed, that as the concentration of ZnO increases, the antimicrobial activity also increases, followed by shear bond strength reduction. It is important to underline, that ZnO and CuO coated brackets have been observed with better antimi-crobial characteristics on Streptococcus mutans than when the brackets were coated with CuO nanoparticles alone [77]. Kachoei et al. [78], Behroozian et al. [79] and Goto et al. [80] proved, that following ZnO nanoparticle coating, the frictional forces be-tween archwires and brackets significantly decreased. Because of that effect, these na-noparticles offer new opportunities in overcoming the unwanted friction forces, better anchorage control, and reduced risk of resorption.

## 5. Relationship between the Orthodontic Arch and Bracket Materials

We use various brackets and arches in orthodontic treatment. The most popular materials from which the locks are stainless steel, titanium, ceramics, and plastic. The materials that arches are usually made of are: Stainless steel, nickel-titanium alloy, chrome-cobalt steel, and titanium–molybdenum alloy. Between the arch and the orthodontic bracket, we can observe the phenomenon of friction, which makes it difficult to move the bracket along the arch. Friction is one of the crucial forces in orthodontics. It acts against the traction force (TF), which can be seen in Figure 3.

The friction observed with orthodontic sliding mechanics is a clinical challenge for orthodontists—the high levels of friction can reduce the effectiveness of the mechanics, reduce the efficiency of tooth movement and further complicate anchorage control [80]. One of the main goals of orthodontic manufacturing companies is to look for new products that would generate less friction during sliding mechanics. One of them is the use of nanoparticles. There are two variables that influence the friction generated during orthodontic treatment: Mechanical and biological [81]. 

Mechanical factors mainly include the material of the bow and bracket. The gold standard of materials for performing sliding is the combination of stainless steel brackets and arches. Based on the research by Kusy and Whitley, the friction force is influenced by the shape and size of the arc. They claim that the friction is greater in larger diameter arches [82]. Several studies show that rectangular wires cause more friction than round wires [83]. The friction also depends on the material of the arc. It has been shown that a SS wire pulled through an SS lock produces the least resistance. NiTi wires produce a little greater friction, while titanium–molybdenum (TMA) alloys the largest (Frank and Nikolai showed that NiTi wire has less friction than SS wire) [84]. Another aspect considered in terms of the friction force is the material of the bracket and the type of the bracket. Kusy et al. [85] compared the friction level of stainless steel and titanium brackets. Titanium showed a greater coefficient of friction. Based on research [86], ceramic brackets produce almost twice as much friction as SS brackets. The new, self-ligating type of brackets appears to cause less friction, but this idea still requires scientific confirmation. 

It appears that the main biological factor influencing friction is the presence of saliva which, depending on the type of bracket and arch, can act as a lubricant or as a “glue”. Its action will therefore increase or decrease friction. Baker investigated the effect of saliva on friction and concluded that human saliva reduces the friction force by 15–19% [87]. The correct composition and amount of saliva are therefore important in maintaining the correct treatment. Debris that can resist on the surface of orthodontic arches also appears to be a significant variable that can increase friction during orthodontic treatment. After 8 weeks of use on orthodontic arches, significant deposits of biofilm were registered. The described nanomaterials affecting the number of bacteria can reduce their number, indirectly affecting the condition of saliva and reducing the amount of plaque on orthodontic elements. Using them could prevent increased frictional forces. 

According to the studies, exposure to the oral cavity for one month can cause a significant slowdown in orthodontic movement (in this case the NiTi arches were tested) due to the accumulation of biofilm [88]. Additionally, the study suggests that the acidic pH produced by the bacteria present in the plaque increases the roughness of the arc and thus the friction between the wire and the bracket [89,90]. One of the ways to create unfavorable conditions for plaque accumulation is to try to include in orthodontic treatment the use of nanoparticles having a proven bacteriostatic effect. Properly-applied particles can also improve the mechanical factors by reducing the friction coefficient at the arc-lock interface.

## 6. Microbial Colonization Associated with Different Kinds of FOAs.

Fixed orthodontic appliances inhibit oral hygiene and create new retentive areas for plaque and debris as can be seen in Figure 4. It could increase the carriage of microbes and subsequent infection and it is one of the common problems that should be avoided in orthodontic treatment.

The most common site for bacterial adhesion and biofilm formation is at the bracket adhesive-enamel junction, an area that is difficult to clean with daily brushing. The plaque that accumulates around orthodontic brackets often results in enamel decalcification, white spot formation, and dental caries adjacent to brackets. It is also difficult to remove microbial growth around orthodontic appliances. Its adherence to the fixed appliance is largely contributed by the bracket material and also the design of orthodontic brackets and ligating method [90,91]. The quantity and the quality of the plaque are influenced by many factors, including surface roughness, and surface-free energy [92]. Electrostatic attractions and van der Waal forces influence the adhesion of microorganisms to surfaces too [93]. Many types of braces are used in orthodontics. Bonded brackets have many advantages over bands such as better aesthetics, ease of placement, and removal and accessibility for oral hygiene [94].

## 7. Introduction of Nanofillers or NP (Silver, TiO_2_) to Orthodontic Adhesives

Orthodontic adhesives showed a higher capacity to retain cariogenic *streptococci* than bracket materials. Previous short-term (24-h) in vitro studies demonstrated comparable or lower and still acceptable shear strength when nano-filled adhesives were used to fix orthodontic brackets.

Compared to traditional orthodontic adhesives, the use of nanofillers reduced the surface roughness of the adhesive; however, this was not true when silver NP was added to this mixture. Nevertheless, evaluation of the long-term effect of nanofiber adhesives on preventing enamel demineralization during orthodontic treatment, particularly around brackets and under orthodontic bands, has not yet been investigated.

Silver has been found to have antimicrobial activity against gram-positive/negative bacteria, fungi, protozoa, some viruses, and strains resistant to antibiotics [95,96], as well as cariogenic *Streptococcus mutans* [97]. Resin composites containing fillers implanted with silver ions had antibacterial properties against oral *streptococci* [22]. The addition of NP silver significantly reduces the adhesion of cariogenic *streptococci* to orthodontic adhesive compared to traditional adhesives, without compromising physical properties (shear bond strength). Adding TiO_2_, SiO_2_, or NP silver to acrylic orthodontic materials’ cold-curing acrylic resins is common during the manufacture of removable orthodontic appliances such as expanders, fixers, and functional appliances which are mainly made of polymethyl methacrylate (PMMA). Compared to natural teeth, bacterial plaque adheres to acrylic resin braces with a larger surface area [98], which may lead to the development of caries-forming flora in the oral cavity. *Candida* Stomatitis is also an inflammation of the oral mucosa characterized by erythema (reddened areas), especially on the palate mucosa [99,100], which sometimes occurs under dentures (denture stomatitis) devices, or fixers.

CA is an opportunistic pathogen, and *Candida* is carried in the oral cavity in 25–75% of the studied populations [101]. A relationship has been suggested between the presence of a removable acrylic apparatus and the *Candida* carrier state, as well as low saliva pH [102]. In one study, the incidence of CA carriers before treatment with removable appliances was 39%; this number increased to 79% after 9 months and after treatment, and decreased to 14% after treatment. Similarly, orthodontic appliances placed on tooth tissues favored a greater proliferation of CA compared to dental appliances. The increase in *Candida* proliferation in people wearing removable appliances is probably due to protection against the natural and mechanical removal of saliva and the defense system [101].

Controlling CA proliferation under removable acrylic appliances can potentially prevent the development of orthodontic stomatitis. It is essential to find alternative therapies to eliminate CA that are tolerant to conventional antifungal drugs [100]. Investigation of the antimicrobial properties of NP acrylic materials and their use in mobile appliances is at an early stage and is limited to in vitro models. Sodagar et al. [54] investigated the changes in the bending strength of PMMA acrylic resin after adding TiO_2_ (0.5%) and SiO_2_ (1%) nanoparticles. The inclusion of NP in acrylic resin adversely affected the flexural strength of the final product and this effect was correlated with the concentration of NP [103]. However, a variable was observed after the addition of silver nanoparticles to the acrylic liquid of the two PMMA resins. 

The mature dental plaque is composed of glucans and various microorganisms, the most common of them is *S. mutans* (the most cariogenic) and *Candida albicans*. Researchers like Shrinivaasan Nambi Rammohan, Ahn, Papaioannou, Fournier, or Brusca explored the relationship between CFUs (*S. mutans* alone, *C. albicans* alone, *S. mutans,* and *C. albicans* in combination) on surfaces of different kinds of orthodontic materials. When *S. mutans* was evaluated alone Shrinivaasan et al. [104], Papaioannou et al. [105], Fournier et al. [106], and Brusca et al. [107] found no obvious difference in the adhesion of *S. mutans* to stainless steel, plastic, and ceramic brackets. Quite different results were obtained Ahn et al. [108] There was a greater number of CFUs on stainless steel brackets than on plastic and ceramic brackets. Titanium and gold brackets showed lesser CFUs than stainless steel brackets. In the case of CA was evaluated alone, titanium brackets had the greatest of CFUs number because of the characteristics rough surface of these brackets [108] and gold brackets had the least number of CFUs because of the inert properties of gold. Plastic and ceramic brackets revealed a greater adherence than stainless steel brackets [107]. When *S*. *mutans* and *C. albicans* were evaluated in combination the clinical situation was different than an individual examination of these microorganisms and showed an antagonistic relationship at least in the initial growth but in the established plaque, they rather seem to exert a synergistic effect. For plastic and ceramic brackets, there was a greater number of CFUs and for metal brackets was the least [107].

Summarizing, titanium had some antibacterial properties but was not effective against the fungi. They grow by hyphae formation and the rough surface helped the increased levels of *C. albicans* [109]. Gold brackets revealed a decreased number of CFUs *S. mutans* and *C. albicans* and it could be inert properties of gold. Plastic and ceramic brackets showed greater levels of CFUs when *C. albicans* were studied alone and in combination with *S. mutans*. On composite yeasts exhibited numerous cell elongations which help in the adhesion mechanism and formation of pseudohyphae. Metallic brackets increase the level of bacterial adhesion compared with ceramic brackets because of the highest critical surface tension (greater surface energy). Stainless steel had an increased potential for microorganism attachment [110]. Properly, the material with high surface free energy will attract more bacteria than material with low surface free energy [110]. 

## 8. Nanomaterials in Orthodontics and Their Use in the Nearest Future

Nanoparticles are increasingly involved in dentistry [111]. They are used more often in conservative dentistry, endodontics [112,113], and prosthetics [111], where they become an integral part of treatment. They are used in irrigating solutions, filling materials and alloy in prosthetics. Their dynamic development should also include other fields of somatology, such as orthodontics. Currently we try find a ways to improve the value of mechanical orthodontic appliances. The use of nanomaterials partially solves this problem. The improvement of the biomechanical value of the orthodontic locks and arches, as well as the interference with the bacterial flora by nanomaterials seem worth developing. In the near future, adding nanoparticles to the materials of appliances will be the gold standard, improving the quality of orthodontic treatment. In addition to determining the basic components of the components of an orthodontic appliance, it will also be necessary to use appropriate proportions of nanoparticles in alloys. In the future, nanoparticles will also partly solve the problem of increased demineralization during treatment, which could reduce the number of complications. The use of the described particles also gives better control of the anchorage. Better control results in better, more predictable treatment [114], which reduces the stress of the orthodontist and increases patient satisfaction [115]. It will be possible to more accurately pursue the goals set at the beginning of treatment—during orthodontic diagnostics. A more thorough treatment will result in a better quality of life for the patient after treatment. The level of compatibility remains a challenge for nanoparticles in the future [116]. Further research is required to determine the safety of their use. Overcoming this problem makes it possible to easily increase the quality of orthodontic treatment. The use of nanoparticles will also reduce the described number of complications during orthodontic treatment, which will result in limiting the performance of additional procedures to eliminate complications. It also reduces treatment time, which reduces the cost of treatment. The shorter treatment time also allows more patients to be healed.

## 9. Materials in Orthodontics and Their Use in the Nearest Future 

The future of nanotechnology in orthodontics has potential to develop in a number of additional applications as well including shape-memory polymers, self-healing materials, self-cleaning materials, biometric adhesives, tooth movement using orthodontic nanobots, and nano-changes on the surfaces of temporary anchorage devices (TADs) to increase their retention but still allow them to be removed when no longer needed [117,118]. 

Shape memory polymers, such as dual shape materials, belong to the group of “actively moving”, which can change shape from one to the other. Orthodontics can use low stiffness transparent polymer arcs that can transform into arcs with a specific modulus of elasticity when exposed to a heat or light for example. With this procedure, it is possible to increase the effectiveness of treatment and aesthetics [119]. Self-healing materials that can repair themselves similar to biological systems. Hybrid materials have been developed, made of micro-ducts containing liquids or dissolved therapeutic agents. These materials can be used in the production of locks and orthodontic arches. A breach of the buckle or wire causes the nanobubble to burst and expose the monomer to the environment, thereby filling the resulting rupture gap with the described therapeutic agents [120].

Biometric adhesives—It is an enamel-friendly bonding mechanism for orthodontic appliances. The process takes place due to the formation of localized van der Waals forces [121]. This action ensures a strong bond between the materials without the use of a chemical substance. This material is often named “geckel”. It acts as like a sticky note and exhibits strong, reversible adhesion in air and in water [122]. In orthodontics, such a procedure would ensure adequate bond strength without prior conditioning of the enamel. Self-cleaning materials have been developed, by using appropriate materials, increase the safety of using orthodontic appliances. The idea was taken from aircraft, where planes are covered with a titanium oxide nanocoating. A super-hybrid layer of hydrofluoric acid forms on the surface to prevent contamination. Photocatalytic activity resulting from the reaction of titanium oxide with light has attracted attention in orthodontic materials [123]. They try to find how inducing a reaction on the alloy of Ni-Ti archwires. By appropriate procedure—thickening the titanium oxide layer, electrolytic treatment and applying heat, it is possible to obtain a crystalline structure of rutile (titanium dioxide) on the surface of the materials [124].

## 10. Conclusions

Nowadays, nanotechnology plays an important role in the dental field since it has the potential to bring significant innovations and benefits. The recent positive results are a stimulus for future research, especially regarding orthodontics. The range of research including orthodontic bonding materials, covering of brackets and wires, as well as their antimicrobial characteristics has a huge potential. The review focused on scientific works concerning the use of nanoparticles in orthodontics that has been published in the literature over the last few years. The physicochemical properties gained by nano-sized materials have augmented the efficiency of orthodontic treatment. In this review, due to indicating the main types of literature reviews and referring to key studies showed that the physicochemical properties gained by nano-sized materials have augmented the efficiency of orthodontic treatment. Information can be implemented by scientists and doctors involved in the orthodontic therapy is included. 

This review has also showed that the nanomaterials application regarding mechanical and antibacterial properties in orthodontics [55]. Nanoparticles can be successfully added to acrylic resins, cements, or orthodontic adhesives to prevent enamel demineralization during orthodontic treatment. Their versatility in clinical orthodontics can be seen in Table 1. 

This review marked, that control and coordinated management of orthodontic treatment is crucial. Dental materials often present limitations during orthodontic treatment, but recently, nanotechnology and science have helped to partially solve some of the limitations. Nanomaterials can successfully reduce friction between the wire and the bracket, which may influence the orthodontic treatment. They are also useful in increasing the antimicrobial characteristics of materials used during treatment. Adding nanoparticles to the adhesives can increase their mean shear bond strength. This review provides several perspectives for the development of use nanomaterials in orthodontic.

Firstly, it is necessary to improve and search for new opportunities in overcoming the unwanted friction forces, better anchorage control, reducing the risk of resorption all should be based on evidence-based medicine and research generating stronger evidence.

Secondly, it is necessary to monitor the treatment of patients who use orthodontic nanomaterials due to the specificity of the oral cavity environment, which is dynamically changing. Biocompatibility and cytotoxicity are important considerations when using new bioactive materials. In the available literature, the knowledge about adverse effects resulting from the use of nanomaterials in orthodontics is limited. Despite the undoubted advantages of nanomaterials, knowledge about them is still incomplete and should be verified and carefully assessed, and the potential benefits should corresponded with the risk.

The application of nanomaterials in dentistry, especially in orthodontics is anticipated to grow further, and an interdisciplinary approach focusing on expertise in dentistry and nanomaterial science is required. The future in orthodontics will benefit greatly through nanotechnology.

## Figures and Tables

**Figure 1 nanomaterials-11-00337-f001:**
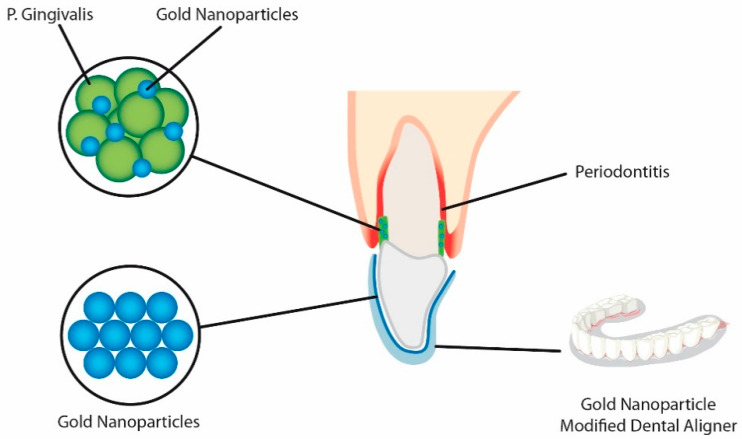
Aligners coated with modified gold nanoparticles causing enhanced antibacterial activity against *Porphyromonas gingivalis*. Due to presence of the coated aligner, the number of bacterial cells was decreased, causing increased biofilm formation prevention.

**Figure 2 nanomaterials-11-00337-f002:**
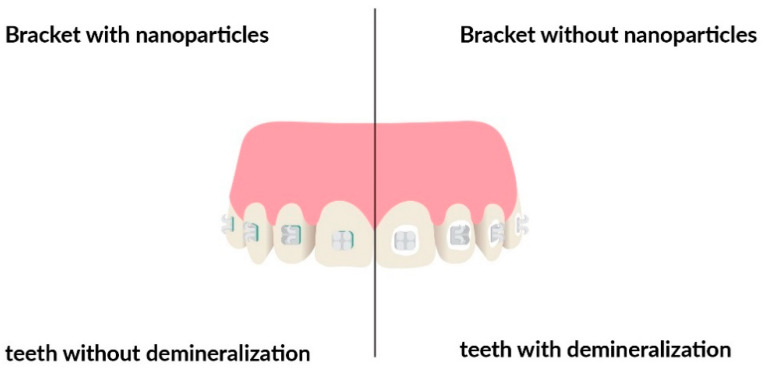
Comparison of teeth demineralization development with and without nanoparticles’ covered brackets.

**Figure 3 nanomaterials-11-00337-f003:**
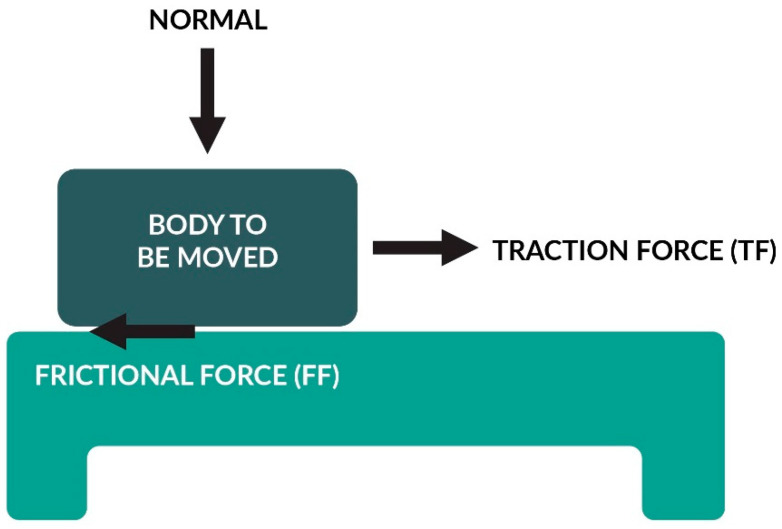
Different forces acting over a body under traction on top of a surface. Body to be moved, traction force (TF), friction force (FF), contact surface (CS).

**Figure 4 nanomaterials-11-00337-f004:**
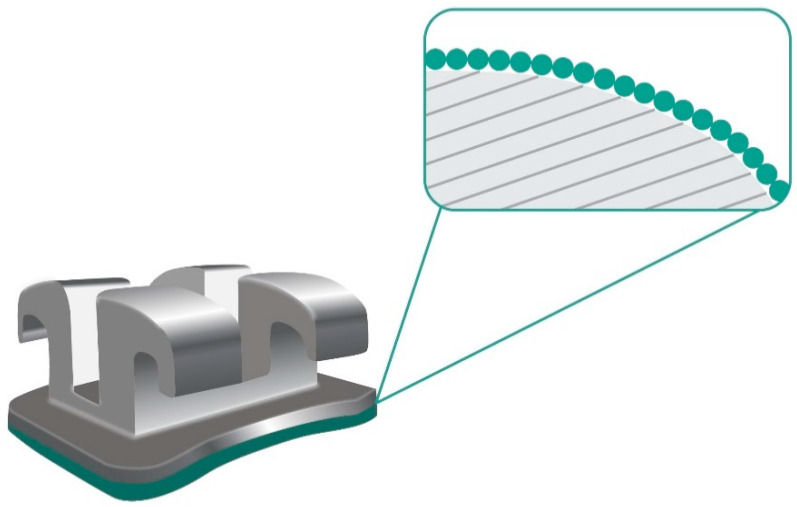
Orthodontic bracket covered by a nano-sized film.

**Table 1 nanomaterials-11-00337-t001:** Nanomaterials application in dentistry.

Nanomaterial	Method of Use	Application	References
Silver NPs (AgNPs)	Applied as a coating agent on titanium	Implants	[40,41]
Zinc oxide NPs (ZnONPs)	Incorporated into dental resins	Resin composite adhesives	[42,43]
Chitosan NPs	Conjugated with silver nanoparticles	Resin composites adhesives	[44,45]
Copper (I) oxide NPs (Cu_2_ONPs)	Antimicrobial effect in resin adhesives	Resin composites adhesives	[46]
Titanium (IV) oxide NPs (TiO_2_NPs)	Nanotubes on titanium surfaces and incorporated with ZnONPs	Implants	[47,48]
Gold NPs (AuNPs)	Modified gold nanoparticles (AuDAPT) coated onto orthodontic aligners	Antimicrobial coated aligner	[49]
Carbonate hydroxyapatite nanocrystal	Antibacterial and antidemineralizing properties	Toothpastes, mouthwashes and composite resins	[50]
Amorphous Calcium Phosphate (ACP)	Antibacterial and antidemineralizing properties	Antibacterial and antidemineralizing properties	[51]
Novel Poly(l-lactic acid) (PLLA)/Multi-walled carbon nanotubes (MWNTs)/hydroxyapatite (HA) nanofibrous scaffolds	Polymer solution FOR entire-tooth regeneration	Dental Surface applications	[52]
Bioactive peptide—Amphiphile nanofibers	Branched peptide Amphiphile molecules containing the peptide motif Arg-Gly-Asp, or “RGD”	Dental surface applications	[53]

## Data Availability

Not applicable.

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
