# Peer review of "Nanomaterials Application in Orthodontics"

_nanomaterials, 2021, doi:10.3390/nano11020337_

Round 1

Reviewer 1 Report

The manuscript sums up the current knowledge about nanomaterials on orthodontic appliances. The novelty of the present review is the focus on nanomaterials.
The present review is interesting and relegant because is go used on nanomaterials applied to operative dentistry field.
The topic is original and relevant. The novelty is the concept of nanotechnology and its applications in the field of orthodontics. Emphasis is put on antimicrobial and mechanical characteristics of nanomaterials during orthodontic treatment.
Paper is well written and designed. Authors should considered to add a paragraph on future perspective.
Text is clear and easy to read.
A paragraph in future perspectives on nanotechologies applied ti orthodontics should be included in the review.

Authors should include a paragraph on new future perspective in the materials used in orthodontics.

Author Response

Dear Reviewer,

We would like to express our sincerest gratitude to the Reviewers for their enormous efforts in criticizing the manuscript. We have taken into account all raised question here follows the detailed answers to the Reviewers. Moreover, all changes we have made to the original manuscript, are marked in the red color in the text.

Review 1

Question: A paragraph in future perspectives on nanotechnologies applied in orthodontics should be included in the review.

Answer: We would like to thank you for the comment. A paragraph in future perspectives on nanotechnologies in orthodontics has been added to the manuscript.

Question: Authors should include a paragraph on new future perspective in the materials used in orthodontics.

Answer: We would like to thank you for the comment. A paragraph in future perspectives on materials in orthodontics has been added to the manuscript.

Reviewer 2 Report

This work is an interesting descriptive review of the literature on the use of nanomaterials in the orthodontic field. Some criticisms are however present: -First of all, the number of authors seems absolutely disproportionate for a review, even descriptive and unsystematic, of the literature. All the work does not follow a correct logical order for which some parts, which have no connection with nanomaterials, are unjustly overexposed and inserted in the discussion on nanos. -Even if it is a descriptive review it is necessary to indicate, at least in summary, the research strategies followed with criteria of inclusion, exclusion, time limits and number of scientific works involved in the work. -Line 28 list in the abstract section the various fields of application of nanomaterials in orthodontics Line31 is missing a final clinical consideration in the abstract section -Line 26 Starting an orthodontic work with the definition of caries seems absolutely inappropriate. Instead, it is necessary to define the main characteristics of an orthodontic treatment. -The introduction section appears extremely confusing. Follow a possible scheme from line 45 onwards: a) definition and classification of nanomaterials b) Their application in general dentistry c) Use in orthodontics d) advantages and disadvantages defined by the literature In this regard, I recommend inserting the following scientific work in the reference section which may be of help to the reader:

Chieruzzi M, Pagano S, Moretti S, Pinna R, Milia E, Torre L, Eramo S. Nanomaterials for Tissue Engineering In Dentistry. Nanomaterials (Basel). 2016 Jul 21; 6 (7): 134. doi: 10.3390 / nano6070134. PMID: 28335262; PMCID: PMC5224610.

- There is also no classification of nanomaterials orthodontic applications -Line 203 the bibliographic reference is missing -All the section of the materials used to build the brakes must be reduced and moved initially while the section that should be the main one, that is, that of nanomaterials, must first be classified (e.g. nanocoating, bracket, adhesives, ectc) and then articulated by paragraphs -In table 1 insert, together with the formulas, also the names of the nanomaterials

Author Response

Dear Reviewer,

We would like to express our sincerest gratitude to the Reviewers for their enormous efforts in criticizing the manuscript. We have taken into account all raised question here follows the detailed answers to the Reviewers. Moreover, all changes we have made to the original manuscript, are marked in the red color in the text.

Review 2

Question: the number of authors seems absolutely disproportionate for a review, even descriptive and unsystematic, of the literature.

Answer: We would like to thank you for the comment. The authors work together in one team, the work is created as part of a student research club under the supervision of experienced clinicians.

Question: All the work does not follow a correct logical order for which some parts, which have no connection with nanomaterials, are unjustly overexposed and inserted in the discussion on nanos. -Even if it is a descriptive review it is necessary to indicate, at least in summary, the research strategies followed with criteria of inclusion, exclusion, time limits and number of scientific works involved in the work.

Answer: We would like to thank you for the comment. Order of Introduction and Conclusion has been changed, additional information about nanomaterials has been added.

Question: Line 28 list in the abstract section the various fields of application of nanomaterials in orthodontics.

Answer: We would like to thank you for the comment. Possible applications of nanomaterials in orthodontics have been added to the manuscript in the Abstract section.

Question: Line31 is missing a final clinical consideration in the abstract section.

Answer: We would like to thank you for the comment. Clinical considerations of nanomaterials have been added to the manuscript.

Question: Line 26 Starting an orthodontic work with the definition of caries seems absolutely inappropriate. Instead, it is necessary to define the main characteristics of an orthodontic treatment.

Answer: We would like to thank you for the comment. The introduction has been prolonged, Additional information about orthodontic treatment has been added to the manuscript.

Question: The introduction section appears extremely confusing. Follow a possible scheme from line 45 onwards: a) definition and classification of nanomaterials b) Their application in general dentistry c) Use in orthodontics d) advantages and disadvantages defined by the literature In this regard, I recommend inserting the following scientific work in the reference section which may be of help to the reader:

Chieruzzi M, Pagano S, Moretti S, Pinna R, Milia E, Torre L, Eramo S. Nanomaterials for Tissue Engineering In Dentistry. Nanomaterials (Basel). 2016 Jul 21; 6 (7): 134. doi: 10.3390 / nano6070134. PMID: 28335262; PMCID: PMC5224610.

Answer: We would like to thank you for the comment. The Introduction section has been changed and corrected. Recommended scientific work has been added to the manuscript.

Question: There is also no classification of nanomaterials orthodontic applications.

Answer: We would like to thank you for the comment. Table 1 concerning nanomaterials use in dentistry has been extended. It also contains possible application of nanomaterials in orthodontics.

Question: Line 203 the bibliographic reference is missing.

Answer: We would like to thank you for the comment. Mentioned part of the manuscript has been corrected.

Question: All the section of the materials used to build the brakes must be reduced and moved initially while the section that should be the main one, that is, that of nanomaterials, must first be classified (e.g. nanocoating, bracket, adhesives, etc.) and then articulated by paragraphs -In table 1 insert, together with the formulas, also the names of the nanomaterials.

Answer: We would like to thank you for the comment. Table 1 has been extended and the names of the nanomaterials have been added to the table. The section about brackets has been reduced and corrected according to the Reviewer’s guidelines.

Reviewer 3 Report

The article is an interesting review for the scientific communit.

It is weel written and structured. I just consider that Table q should be enlarge includig more information, such as more particles types and information about the particles sucha as diameter/size and methods used to prepare them

Author Response

Dear Reviewer,

We would like to express our sincerest gratitude to the Reviewers for their enormous efforts in criticizing the manuscript. We have taken into account all raised question here follows the detailed answers to the Reviewers. Moreover, all changes we have made to the original manuscript, are marked in the red color in the text.

Review 3

The article is an interesting review for the scientific community.

It is well written and structured. I just consider that Table q should be enlarge including more information, such as more particles types and information about the particles such as diameter/size and methods used to prepare them.

Answer: We would like to thank you for the comment. The table has been enlarged and now contains more nanomaterials with references.

Round 2

Reviewer 2 Report

I reccomend work acceptation